# Semantic Mosaicing of Histo-Pathology Image Fragments using Visual Foundation Models

**Stefan Brandstätter**[1,3,4]          STEFAN.BRANDSTAETTER@MEDUNIWIEN.AC.AT

**Maximilan Köller**[5]          MAXIMILIAN.KOELLER@MEDUNIWIEN.AC.AT

**Philipp Seeböck**[1,3,4]          PHILIPP.SEEBOECK@MEDUNIWIEN.AC.AT

**Alissa Blessing**[5]          ALISSA.BLESSING@MEDUNIWIEN.AC.AT

**Felicitas Oberndorfer**[3,5]          FELICITAS.OBERNDORFER@MEDUNIWIEN.AC.AT

**Svitlana Pochepnia**[2,3]          SVITLANA.POCHEPNIA@MEDUNIWIEN.AC.AT

**Helmut Prosch**[2,3]          HELMUT.PROSCH@MEDUNIWIEN.AC.AT

**Georg Langs**[1,3,4]          GEORG.LANGS@MEDUNIWIEN.AC.AT

[1]*Computational Imaging Research Lab, Dept. of Biomedical Imaging and Image-guided Therapy*

[2]*Division of General and Pediatric Radiology,Dept.of Biomedical Imaging and Image-guided therapy*

[3]*Christian Doppler Laboratory for Machine Learning Driven Precision Imaging, Department of Biomedical Imaging and Image-guided therapy*

[4]*Comprehensive Center for Artificial Intelligence in Medicine*

[5]*Department of Pathology*

*Medical University Vienna, Vienna, Austria*

## Abstract

In histopathology, tissue samples are often larger than a standard microscope slide, making stitching of multiple fragments necessary to process entire structures such as tumors. Automated stitching is a prerequisite for scaling analysis, but is challenging due to possible tissue loss during preparation, inhomogeneous morphological distortion, staining inconsistencies, missing regions due to misalignment on the slide, or frayed tissue edges. This limits state-of-the-art stitching methods using boundary shape matching algorithms to reconstruct artificial whole mount slides (WMS). Here, we introduce SemanticStitcher using latent feature representations derived from a visual histopathology foundation model to identify neighboring areas in different fragments. Robust pose estimation based on a large number of semantic matching candidates derives a mosaic of multiple fragments to form the WMS. Experiments on three different histopathology datasets demonstrate that SemanticStitcher yields robust WMS mosaicing and consistently outperforms the state of the art in correct boundary matches.

**Keywords:** Whole-Mount Sectioning (WMS), UNI, Histopathology, Image Stitching, Foundation Model

## 1 Introduction

While microscope slides are essential for pathology, their size limits full specimen analysis. (Duan et al., 2024) Whole-Mount histopathology addresses this but introduces new scanning challenges. Artificial WMS mosaicing offers a solution through fragment alignment. Whole-Mount Histopathology (WMH) is a comprehensive technique examining the entire cross-section of a specimen resulting in a Whole-Mount Sectioning (WMS) large-format

slide. (Cimadamore et al., 2020) It captures the full spatial distribution and morphological features of tissue relevant for diagnosis and research. WMS enhances histopathology-imaging correlation, reduces cutting artifacts, and preserves tissue context. (Schouten et al., 2024) However, it also introduces challenges, including the need for larger, more costly scanners, and technical limitations in capturing oversized slides. (Duan et al., 2024) To address these limitations, obtaining the advantages of WMS with standardized image acquisition protocols, ongoing research focuses on creating artificial WMS images by aligning tissue fragments. (Chappelow et al., 2011; Penzias et al., 2016; Schouten et al., 2024)

**Related work**   Several stitching methods have been proposed. HistoStitcher (Chappelow et al., 2011) requires manual landmark selection and transformation tuning, making it too labor-intensive for clinical use. AutoStitcher (Penzias et al., 2016) was the first fully automated method, using L2-norm histogram differences with a misalignment term, but still requiring manual fragment labels. PythoStitcher (Schouten et al., 2024) is the current state-of-the-art, applying a boundary-based, multi-resolution strategy for high-resolution mosaicing without manual input or extra cost functions. It performs poorly on irregularly shaped or equally sized boundary fragments. To address these limitations, we propose a novel method for automatic mosaicing of artificial WMS from given tissue fragments.

**Contribution**   We introduce SemanticStitcher, an automated WMS mosaicing method that aligns sets of arbitrarily shaped tissue fragments by matching and integrating their visual content. Unlike state-of-the-art boundary-based algorithms, it requires no prior knowledge of tissue shape, arrangement, or fragment count. Instead, it leverages semantic features extracted with the help of a foundation model to effectively compare the content of image patches at high resolution. It uses the resulting similarities to perform robust and accurate alignment of fragment boundaries yielding an artificial WMS from a set of segments.

## 2 Method

*SemanticStitcher* accurately aligns sets of digitized tissue fragments $\mathbf{X} \in \mathbb{R}^{h \times w}$ to reconstruct an artificial WMS (Fig. 1). The algorithm selects a fragment from a fragment pool, pairs it with its best-matching counterpart, and computes the rotation matrix $\mathbf{R}$ and the translation vector $\mathbf{t}$ required for precise alignment of the fragment. It then iteratively stitches subsequent fragments to the accumulating mosaic to finally yield an artificial WMS containing all fragments. The approach consists of two stages: (1) Identifying neighboring fragment pairs and corresponding semantic match candidates and (2) robustly estimating their spatial alignment for mosaicing.

**Stage 1: Fragment pairing**   As a simple preprocessing step, for each fragment $\mathbf{X}$ in the fragment pool we remove the background (OTSU (Otsu, 1979)) and detect the boundary $\mathcal{B}_X$ (Suzuki and be, 1985). Along all the fragment boundaries, patches $\mathbf{P}_X^{(k)} \in \mathbb{R}^{ph \times pw}$ are sampled at fixed intervals. Each patch is encoded into a feature vector $\mathbf{S} : \mathbf{P}_X^{(k)} \mapsto \mathbf{f}_X^{(k)} \in \mathbb{R}^K$ using a pre-trained semantic encoding model. In our experiments we evaluate the UNI and CONCH foundation model for this purpose (Chen et al., 2024; Lu et al., 2023). A random fragment $\mathbf{X}_M$ out of the pool is selected as the moving fragment. All other fragments are treated as fixed fragments $\mathbf{X}_F$. To determine the best fragment match, we compute the cosine similarity between all boundary patch feature vectors of the moving fragment

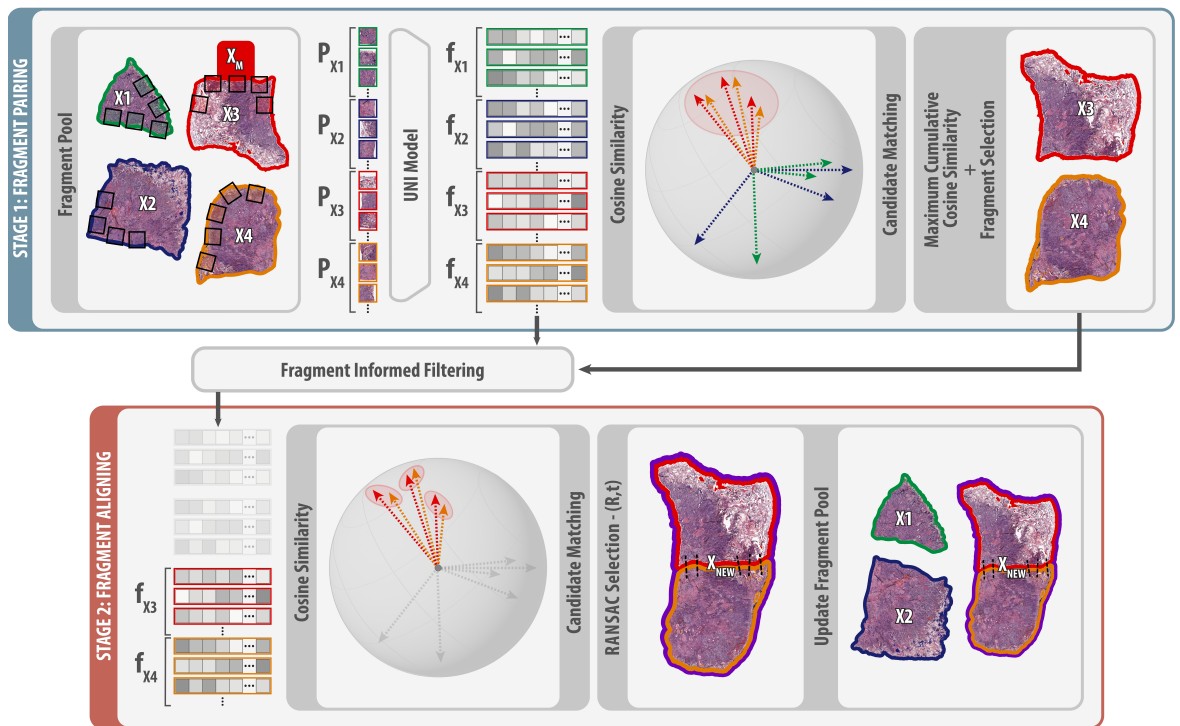

Figure 1: **SemanticStitcher:** In Stage 1, a fragment is randomly selected from the fragment pool and paired with its most compatible counterpart. In Stage 2, the optimal transformation aligning the paired fragments is computed.

$\mathbf{X}_M$ and all boundary patch feature vectors of all fixed fragments $\mathbf{X}_F$. For each feature vector in the moving fragment $\mathbf{f}_{X_M}^{(k)}$, we identify the most similar feature vector on the fixed fragments $\mathbf{f}_{X_F}^{(j)}$, forming candidate matches. To select a matching fragment, we sum up the cosine similarity for each $\mathbf{X}_F$ over all the boundary encodings and select the fixed fragment with the highest score as the optimal match for $\mathbf{X}_M$.

**Stage 2: Fragment alignment** After selecting a pair of fragments, we perform alignment, by first restricting comparison of feature vectors to those of the matched fragment pair $(\mathbf{X}_M, \mathbf{X}_F)$ from Stage 1. We then identify candidate matches between $\mathbf{X}_M$ and $\mathbf{X}_F$ by finding feature pairs with the highest similarity. These candidate matches form the basis for a robust estimation of alignment parameters between the fragments using RANSAC (Fischler and Bolles, 1981). It mitigates the impact of outliers and incorrectly matched pairs, and identifies the pose parameters consistent with a majority of matches. Once $\mathbf{X}_M$ is aligned with $\mathbf{X}_F$, the resulting composite fragment replaces the two initial fragments of the pair in the fragment pool, and the fragment selection and alignment is repeated. This iterative process continues until a single mosaic fragment remains, forming the complete artificial WMS. In the following we provide details of the individual steps of the approach.

**Patch extraction** For patch extraction, in our experiments, we begin with a boundary point and identify the next boundary point at a distance of 224 pixels, corresponding to

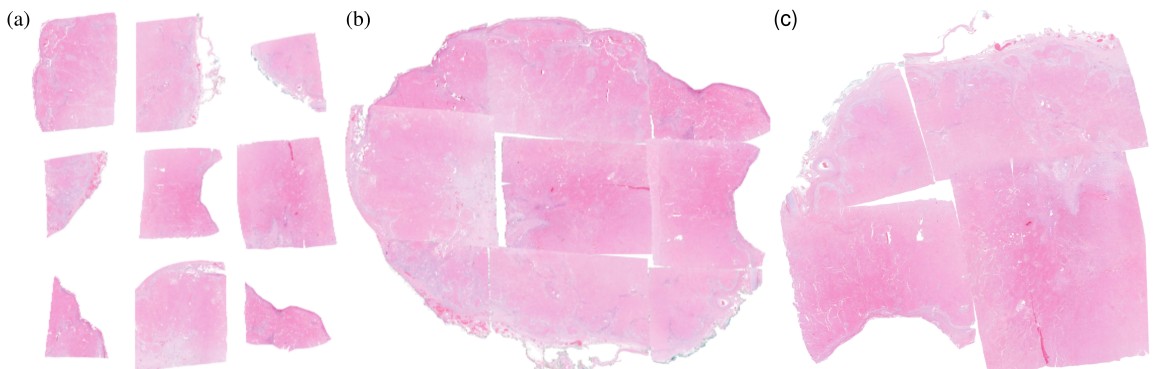

Figure 2: Experiment A: (a) Visualization of raw fragments without preprocessing, (b) WMS reconstruction using SemanticStitcher, (c) WMS reconstruction using the state-of-the-art PythoStitcher algorithm

the input size of the UNI encoder. The patch $\mathbf{P}_X^{(k)}$ is then extracted orthogonal to the line segment connecting the two points, ensuring the patch orientation is perpendicular to their spatial alignment. To ensure comprehensive coverage of the boundary $\mathcal{B}_X$, patches are extracted with an overlap of half the patch size. Patch extraction is performed along the entire boundary but shifted 10 pixels inward to avoid black pixels from frayed edges.

**Patch encodings** To encode the patches $\mathbf{P}_X^{(k)}$ into meaningful feature vector representations $\mathbf{f}_X^{(k)}$, we utilize the final layer of the state-of-the-art foundation model UNI, a general-purpose, self-supervised model for pathology. (Chen et al., 2024) We follow the preprocessing described in the UNI paper.

**Feature vector matching** To establish correspondences between the moving fragment $\mathbf{X}_M$ and fixed fragment $\mathbf{X}_F$, we consider both local patch correspondences and broader spatial relationships. For each patch $\mathbf{P}_X^{(k)}$, we extract its associated feature vector $\mathbf{f}_X^{(k)}$ along with three preceding and three following feature vectors (neighbourhood 3), forming a context-aware feature stack $\mathbf{F}_X^{(k)} = [\mathbf{f}_X^{(k-3)}, \ldots, \mathbf{f}_X^{(k)}, \ldots, \mathbf{f}_X^{(k+3)}]$. We then compute the cosine similarity between each stacked feature vector $\mathbf{F}_{X_M}^{(k)}$ from $\mathbf{X}_M$ against candidate stacks $\mathbf{F}_{X_F}^{(j)}$ from $\mathbf{X}_F$ using a sliding window approach. The highest-scoring pair for each $\mathbf{F}_{X_M}^{(k)}$ is selected, generating candidate matches.

**Robust Transformation Estimation** We use the candidate matches to calculate the transformation between the moving and fixed image using the RANSAC algorithm. RANSAC provides the most consistent rotation matrix ($\mathbf{R}$) and translation vector ($\mathbf{t}$), both of which are utilized for the final alignment of the moving and fixed image.

## 3 Experimental Setup

**Data** We utilized three medical imaging datasets containing data of two different organs. The first dataset, TCGA-LUAD (Albertina et al., 2016), comprises 514 tissue slides of lung adenocarcinoma. The second dataset, TCGA-PRAD (Zuley et al., 2016), consists of 490 tis-

Table 1: Quantitative analysis of boundary matches (in %).

| Method | TCGA-LUAD | TCGA-PRAD | IN-HOUSE |
|---|---|---|---|
| | | Matches in % ↑ | |
| PythoStitcher (Schouten et al., 2024) | 42.21 | 46.12 | 38.88 |
| **SemanticStitcher (ours)** | **81.33** | **76.05** | **86.11** |

sue slides of prostate adenocarcinoma. Both datasets were reduced to 310 and 254 samples, respectively, due to the presence of painted slides, insufficient resolution, and excessively frayed or torn tissue samples, rendering them unsuitable for analysis. These samples were filtered out as part of our preprocessing pipeline. We use the datasets to artificially simulate different fragment arrangements, and evaluate the corresponding performance of the stitching algorithm. The third dataset is an in-house set comprising 8 hematoxylin and eosin (HE) stained lung cancer specimens, scanned using an Olympus VS200 slide scanner, at resolution 0,274 µm/px. This dataset comprises real fragments with corresponding irregularities representative of clinical procedures. We performed 5 evaluation experiments (A-E). Experiment A was performed on the in-house dataset, Experiment B on all three datasets and Experiment C, D and E using the TCGA-LUAD and TCGA-PRAD datasets. **Implementation details** All slides were processed at a resolution of 1 µm and reconstructed at 0.25 µm. For all analyses, we used the pretrained UNI model. Additionally, for experiment B, we incorporated the pretrained CONCH model (Lu et al., 2023). For the UNI/CONCH model the extracted patches $\mathbf{P}_X^{(k)}$ had a size of $224{\times}224/448{\times}448$ and the feature vectors $\mathbf{f}_X^{(k)} \in \mathbb{R}^K$ had a size of $K = 1024/K = 768$. For RANSAC (Fischler and Bolles, 1981) we chose an inlier threshold of 500 pixel, a maximum of 1000 iterations, and a minimum of 6 points required for model estimation.

**Experiment A: Artificial WMS reconstruction in clinical practice** We assess the applicability of our method in real clinical practice data on our in-house dataset of histological fragments generated during routine diagnosis. These fragments were reconstructed into an artificial WMS, and the arrangement was compared to the expert-provided ground truth arrangement. We compared SemanticStitcher to the state of the art PythoStitcher. **Experiment B: Quantitative evaluation of tissue alignment** We quantitatively evaluate the tissue alignment accuracy on all three datasets by (1) artificially splitting a tissue slide into fragments, (2) mosaicing the artificial fragments to a WMS, and (3) counting the number of correctly vs. incorrectly matched boundaries. **Experiment C: Spatial awareness and impact of RANSAC** We visualize the connections between patches in the moving fragment and their corresponding matches in the fixed fragment before and after RANSAC. Additionally, we qualitatively assess spatial and semantic awareness - or capture range - of the feature space by visualizing the cosine similarity between an encoded patch from the image center and all other tissue patches. **Experiment D: Accuracy of correct match prediction with increasing gap size** We evaluate patch matching accuracy before and after RANSAC by artificially splitting a tissue slide into two fragments, and progressively increasing the gap between them before matching. This simulates real-world conditions, where tissue gaps may be larger due to morphological distortions, misalignments on the slide, or inaccurate cuts during routine processing. We assess how patch distance affects feature embeddings via cosine similarity between patches with an offset varying between

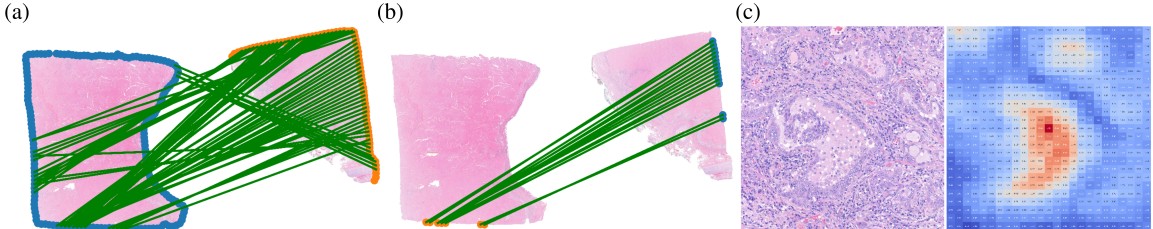

Figure 3: Experiment C: (a) Visualization of all semantic match candidates between two matched fragments; (b) valid connections retained after RANSAC filtering; (c) a tissue slide (left) and the feature space cosine similarity between the central patch and the rest of the image (right).

0 and 550 $\mu m$. **Experiment E: Rotation invariance, neighborhood analysis and resolution invariance** We assessed the practicality of our method by conducting in depth analyses of (1) rotation invariance, to ensure that a rotated patch remains most similar to itself, (2) the effect of neighborhood size on feature matching, and (3) the trade-off between resolution and computational speed.

## 4 Results

**A.** Fig. 2 shows qualitative results of real-world fragment mosaicing without any predefined orientation or arrangement. SemanticStitcher yields robust mosaicing results with all boundaries correctly matched, and only minor alignment errors were observed in fragments with frayed boundaries (b top right segment) or where portions of the tissue were missing from the slide during imaging (b center segment). In contrast, we observed significant inaccuracies in fragment positioning and stitching edge alignment for the state-of-the-art boundary-based approach. It has difficulty processing fragments with similar-length boundaries (squares), as observed in (c) at the bottom left and right.

**B.** Table 1 reports the ratio of correct boundary matches in %. Results demonstrate that SemanticStitcher consistently outperforms the state-of-the-art algorithm across all three datasets by a large margin. For the in-house dataset, no preprocessing was applied, and both algorithms received raw fragments. For the TCGA-LUAD and TCGA-PRAD datasets, tissue slides were split into four segments of approximately the same size to facilitate boundary matching. To simulate routinely scanned slides, we increased the gap between fragments to the size of one patch and randomly reduced the stitching edges by 0–20% to mimic variable boundary lengths.

**C.** Fig. 3(a) displays a typical example of fragment matching (Stage 2), with all matched patches connected by lines. Several incorrect matches are primarily due to factors such as staining variations, morphological distortions (e.g., tissue shrinkage) and frayed tissue edges introducing black pixels during encoding. Fig. 3(b) illustrates the removal of mismatches by applying the RANSAC algorithm. This is in line with quantitative results, demonstrating that erroneous connections can be effectively discarded by a consensus operation across patches, omitting the need for perfect matches for every patch. Fig. 3(c) illustrates the

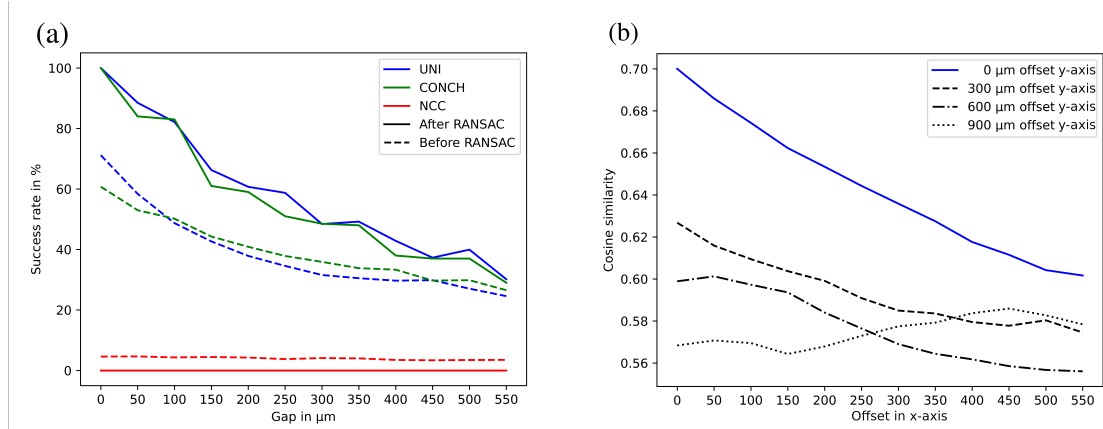

Figure 4: Experiment D: (a) Fraction of correct patch matches before and after RANSAC for three different models (UNI, CONCH, NCC). (b) Cosine similarity of patch pair encodings with spatial separation to simulate positional offset. (a) and (b) are evaluated with increasing gap.

cosine similarity within the feature space by comparing the central patch's feature vector to those of all other patches in the image. The similarity visually reflects the tissue type structure, suggesting a feature space representing relevant information.

**D.** Fig. 4 (a) shows correct patch matches as percentage of all matches, before and after RANSAC, for three similarity measures: semantic features extracted with the UNI model, with the previously published model CONCH, and with a standard patch-level normalized-cross-correlation-based similarity (NCC) of the image pixel values. Evaluation was performed for different gap sizes. Matches were compared to the known ground-truth correct match. Even at a 250 $\mu m$ gap—larger than a patch—SemanticStitcher matched over 40% of cases without RANSAC. Fig. 4 (b) demonstrates that the cosine similarity of nearby patches is higher compared to patches further away by comparing the similarity resulting from different offsets. This indicates that using cosine similarity of feature embeddings both encodes spatial and semantic information.

**E.** Fig. 5 (a) demonstrates the robustness of the similarity between patches in face of patch. The rotated patch remains more similar to its non-rotated version compared to four neighboring patches (top, right, left, and bottom). Fig. 5 (b) illustrates the relevance of matching sets of patches over single patches. The success rate of correct matches based on cosine similarity increases significantly from 30% to 90% when expanding from single patch to a neighborhood size of 3 Fig. 5 (b). We evaluated SemanticStitcher across different resolutions ranging from 0.25 to 4 $\mu m$ and identified 1 $\mu m$ as the optimal balance between efficiency and resolution.

## 5 Discussion and Conclusion

SemanticStitcher is a fully automated method for artificial WMS mosaicing from sets of fragments. It aligns scanned fragments using semantic matching and robust pose estima-

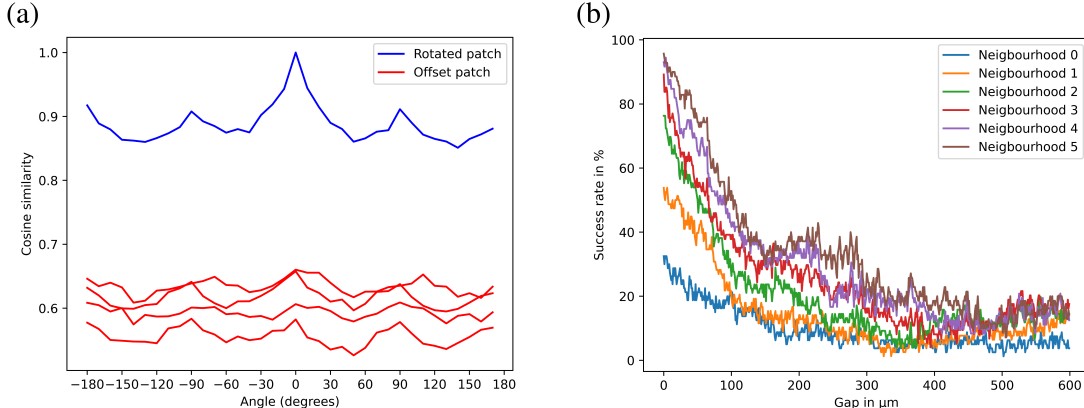

(a)

(b)

Figure 5: Experiment E: (a) Similarity between a rotated patch and its correct non-rotated counterpart (blue) compared to neighboring patches (red) illustrating the robustness of the comparison to rotations. (b) Success rate for matched patches for different neighborhood sizes as the gap size increases.

tion. Evaluation on three datasets shows its effectiveness on both fragments simulated from a WMS and on real clinical fragment data. Unlike in purely boundary-based methods, high-resolution semantic similarity enables more accurate and robust mosaicing. It captures neighborhood relationships despite boundary distortions from image acquisition. Compared to cross-correlation between pixel values, image embeddings by foundation models offer more stable representations for alignment. Boundary-based methods struggle with irregular or equally sized fragments because they rely on matching boundary lengths. Though currently evaluated only on HE-stained slides, it is expected to generalize to other stainings with suitable embedding models. By handling boundary irregularities and diverse fragment layouts, SemanticStitcher can streamline workflows and enhance histopathology analysis.

## Acknowledgments and Disclosure of Funding

The financial support by the Austrian Federal Ministry for Digital and Economic Affairs, the National Foundation for Research, Technology and Development and the Christian Doppler Research Association, Siemens Healthineers, the Austrian Science Fund (FWF, P 35189-B - ONSET), the Vienna Science and Technology Fund (WWTF, PREDICTOME [10.47379/LS20065]), and the European Union's Horizon Europe research and innovation programme under grant agreement No.101100633— EUCAIM are gratefully acknowledged.

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
