# OpenReview forum: "Semantic Mosaicing of Histo-Pathology Image Fragments using Visual Foundation Models"
_MICCAI.org/2025/Workshop/COMPAYL — COMPAYL 2025_

### Official Review · Reviewer_PVb9 · 2025-07-15
**Interesting application and techniques**

**Rating:** 4
**Confidence:** 4

**Review:**

The paper presents an interesting idea of virtually connecting tissue parts to a virtual single slide. In my experience usually larger slides can be used but these can indeed cause challenges for whole slide scanners, so even though the scenario may not be the most frequent, this is an interesting application fr techniques.
The described techniques make sense and include novelty over existing mechanisms for such stitching.
Most data sets are openly accessible but do not contain tissue that was created in such a cut way, only the very small private data set is really highlighting the scenario. In the other data tissues are digitally cut, which will likely create pretty different images than when tissue is fixed on several slides, particularly when looking at the techniques. When stained separately it is less likely that adjacent tissue parts contain extremely similar features whereas this is clearly the case when virtually cut apart.
It is not easy to find strong baselines for this and the authors make an effort using existing techniques and also several foundation models.
Resolution is at 1 um/pixel, so approximately 10x magnification. Most foundation models would rather be rained on 20x, I would say.
"correct boundary" is a bit of  a strange evaluation but I agree that this can make sense and that there is nothing else that is obvious for such an evaluation.
"Representative case" should rather be "typical case" because it is likely not really representative from a statistics viewpoint.
Overall an interesting paper that would be interesting to have as a presentation at the workshop, I find. There is novelty and an evaluation with basic baselines.

---

### Official Review · Reviewer_TnZB · 2025-07-15
**Promising semantic stitching approach leverages foundation models effectively, with opportunity for expanded clinical validation**

**Rating:** 4
**Confidence:** 2

**Review:**

Contribution:
The authors propose a new method to tackle the problem of stitching Whole-slide images into Whole-mount slides.  Often, a large piece of tissue is sectioned and each section is then individually images. The separate processing of these sections comes with some important downsides for subsequent computational processing and therefore stitching the images together should provide a more complete view of the subject. The authors propose SemanticStitcher which uses Foundation Models to extract patch-level features that are used to conditionally find a suitable stitching configuration automatically. The method is compared to PythoStitcher, the current state-of-the-art in WMS stitching. The quantitative analyses on two TCGA datasets and one in-house dataset show greatly improved results, albeit this experimental setting does not reflect a realistic, or perhaps clinically relevant scenario. The other experiments validate the robustness of the model in various settings, which are interesting empirical findings that can inform future work, yet an expanded discussion of the methodological and experimental setting would have been a welcome addition.

Strengths:
- The introduction and background section contains most essential information to understand the problem and the current state of research.
- The proposed method uses out-of-the box solution using publicly available foundation models, which reduces the implementation effort (no supervised training or labeling needed) and increases reproducibility. The method used is also intuitive to understand and shows great potential.
- Quantitative analysis on two publicly available datasets, making it reproducible.
- Insightful additional experiments to show the robustness of the proposed solutions. Experiment D, where the method is evaluated on increasing gap sizes somewhat simulates more realistic scenarios and therefore provides interesting empirical results. Comparing SemanticStitcher to PythoStitcher would have been interesting in this scenario. Experiment E, where rotation invariance and neighborhood incorporation are analyzed are also very relevant.

Weaknesses:
- Since the TCGA datasets were not originally intended for whole-mount stitching, it remains unclear whether performance on artificially sectioned whole-mount slides translates to real clinical scenarios. The clinical example of Experiment A does point in that direction, which is comforting, yet more clinical validation (e.g. a reader study) would have convinced the reviewer more. The lack of evaluation on realistic scenarios represents the main weakness of the current paper, especially since Schouten et al. (2024) already laid the groundwork for careful, clinical analysis of their stitching algorithm.
- The paper lacks a critical discussion of the limitations of the proposed method and experimental design, as well as proposals for future research, except for mentioning the applicability of the method on differently stained images.
- In Figure 5b it is unclear to me what the neighborhoods represent. I believe each line shows the inclusion of more neighborhood tiles in the stacked feature vector, but does ‘neighborhood 5’ mean 5 extra neighboring tiles (how were they selected)? Or does it mean something else entirely? Phrasing this differently in the figure or the caption for the camera ready version would be recommended.
- Source code is not released (while PythoStitcher is), reducing reproducibility. If it will be made available, an added footnote with the URI in the camera ready version would be recommended.

---

### Official Review · Reviewer_24XD · 2025-07-15
**This manuscript introduces SemanticStitcher, a fully automated approach for mosaicing whole slide images (WSIs) from histopathology fragments using high-resolution semantic features and robust pose estimation. The method is designed to overcome the limitations of existing boundary-based stitching algorithms, particularly in handling fragments with irregular shapes or ambiguous boundaries. The authors demonstrate that SemanticStitcher outperforms state-of-the-art methods on both synthetic and clinical datasets, offering practical benefits for digital pathology workflows. While the approach is promising, several methodological details require clarification. In particular, more explanation is needed regarding patch sampling strategies, the specifics of similarity computation and aggregation, and the dataset preprocessing steps. Minor writing, terminology, and figure presentation improvements would also enhance clarity. Overall, the paper makes a significant contribution to computational pathology, though greater transparency in the methods and further discussion of failure cases would strengthen the work.**

**Rating:** 4
**Confidence:** 4

**Review:**

Major Comments:
The manuscript notes that PythoStitcher performs poorly on irregularly shaped or equally sized boundary fragments. It would strengthen the work to elaborate on the underlying reasons—for example, does the method depend on consistent boundary geometries, or is it unable to resolve ambiguities in symmetric edges?

In the sentence “Along the fragment boundaries, patches Px are sampled at fixed intervals,” please clarify whether sampling occurs along all boundary edges of the tissue fragments, or only a subset.


Regarding the similarity computation step: it is stated that cosine similarity is computed between all feature vectors of the moving fragment XM and the boundary patches of fixed fragments. Please specify whether all feature vectors of XM are used, or only those near the boundary.

The explanation “we sum up the cosine similarity for each XF and select the fixed fragment with the highest score as the…” should be more precise. Please clarify over what exactly the cosine similarities are summed.

It is mentioned that TCGA-PRAD and TCGA-LUAD were reduced due to painted slides, insufficient resolution, and frayed tissue. Please specify whether the authors performed this filtering as part of preprocessing, or if the datasets already excluded such samples.

The sentence “The patch PX(k) is then extracted perpendicular to the line connecting these two points” would benefit from additional clarification.

Minor Comments:
In the sentence “…making it too labor-intensive for clinical use AutoStitcher (Penzias et al., 2016),” a period is missing. Adding a full stop after “clinical use” would improve readability.

The phrase “direct image content stitching” in “SemanticStitcher…using direct image content stitching” could be reworded for clarity.

For Figure 3(a), using a distinct color or dashed lines to indicate incorrect or mismatched fragment connections would make failure cases easier to distinguish from correct matches.

Overall strengths:
This paper presents SemanticStitcher, a fully automated method for whole slide image (WSI) mosaicing that aligns histology fragments using high-resolution semantic similarity and robust pose estimation. The method eliminates the need for manual intervention, offering a practical solution for clinical workflows. It shows strong performance on both synthetic and clinical datasets, with clear improvements over prior methods—particularly in handling arbitrary fragment configurations. This significantly contributes to computational pathology and advances automated digital pathology pipelines.

Overall weaknesses:
Although the paper introduces a strong method, some technical explanations would benefit from additional clarity, especially regarding patch sampling, similarity computation, and fragment matching procedures. A few improvements in writing and visualization would also enhance the manuscript’s accessibility and interpretability. Additionally, a deeper discussion of failure cases (such as with irregularly shaped fragments) would further strengthen the evaluation.